# Heteroepitaxial Growth of InBi(001)

**DOI:** 10.3390/molecules29122825

**Published:** 2024-06-13

**Authors:** Thomas J. Rehaag, Gavin R. Bell

**Affiliations:** Department of Physics, University of Warwick, Coventry CV4 7AL, UK; thomas.rehaag@warwick.ac.uk

**Keywords:** topological semimetal, III–V semiconductor, molecular beam epitaxy, InBi, InSb

## Abstract

InBi is a topological nodal line semimetal with strong spin–orbit coupling. It is epitaxially compatible with III–V semiconductors and, hence, an attractive material for topological spintronics. However, growth by molecular beam epitaxy (MBE) is challenging owing to the low melting point of InBi and the tendency to form droplets. We investigate approaches for epitaxial growth of InBi films on InSb(001) substrates using MBE and periodic supply epitaxy (PSE). It was not possible to achieve planar, stoichiometric InBi heteroepitaxy using MBE growth over the parameter space explored. However, pseudomorphic growth of ultra-thin InBi(001) layers could be achieved by PSE on InSb(001). A remarkable change to the in-plane epitaxial orientation is observed.

## 1. Introduction

Topological semimetals have revealed a wide range of fascinating electronic and spintronic properties over the last decade [1,2,3,4,5,6]. These include spin–momentum locking and a lack of backscattering (dissipationless transport) in topologically non-trivial surface and interface states. The role of topology as an exciting new aspect of condensed matter physics is firmly established [7,8], and the topic has deep links to other aspects of fundamental physics [9,10,11]. However, numerous barriers remain for exploiting the effects of topology in novel device structures. The remarkable spin-polarized band structures of topological materials could be exploited for low-power spintronic memory and processing architectures. This goal is particularly important given the rapidly increasing energy demands of data centers, which use silicon CMOS technology and magnetic mass storage media, for which almost half the energy consumption is due to cooling [12,13].

One of the first families of topological semimetals to be proposed was the alkali metal bismides [14], and the electronic structure [6] and chiral anomaly [15] were confirmed in Na_3_Bi. Subsequently, electrical control of the topological phase in ultra-thin Na_3_Bi was demonstrated using a scanning tunneling microscope (STM) to apply the electric field [16]. This opens up the possibility of highly efficient “topological transistor” devices [17]. However, the material requirements for scaling the fabrication of such devices beyond the demonstration of the physical principles are strict.

Although topological materials are rather common [18], it is likely that a much smaller subset will be useful in emerging technologies. In order to exploit topological properties in spintronic devices, a huge advantage is compatibility with existing semiconductor materials. This allows established engineering practices to be brought to bear, simplifies the interconnection of topological and regular materials, and enhances future scalability. Another advantage of epitaxial growth of topological semimetals is the ability to tune electrical properties by film thickness, where both quantum confinement and coupling of upper and lower interface states strongly affect the electronic structure [19]. Many topological materials include a group-15 element, for example, Cd_3_As_2_ [20,21], suggesting that the III–V semiconductors could be compatible. Full *epitaxial compatibility* is highly desirable, meaning both that the elements involved are compatible with a III–V growth technology such as molecular beam epitaxy (MBE) and that a suitable epitaxial match of crystal symmetries can be found.

An example of an emerging epitaxially compatible topological semimetal is TaAs [5,22]. This has been successfully grown by MBE on GaAs substrates [23,24,25,26]. The refractory metal Ta can be readily controlled in an MBE environment without introducing contamination (e.g., leading to unintentional doping of semiconductor layers). The body-centered tetragonal crystal structure of TaAs has a simple epitaxial relationship with the (001) faces of cubic III–V semiconductors, although the lattice mismatches are substantial. SrMnSb_2_ is another topological semimetal with potential compatibility with III–V semiconductors. To date, it has been studied only in bulk form [27,28,29], but we have recently shown that MBE growth on III–V semiconductors is possible in a similar manner as the ferromagnetic weak metal MnSb [30,31]. The ultrafast properties of topological semimetals should also be highlighted [32,33,34]. Excitation by laser light pulses can allow the topological state to be changed at ultrafast timescales via excitation of coherent phonons. Both TaAs and SrMnSb_2_ exhibit coherent phonons [35,36], and such properties could be enhanced by epitaxial control of film thicknesses and the phononic properties of the surrounding material.

InBi is a topological nodal line semimetal [37] with a III–V-like stoichiometric composition. It is epitaxially compatible with III–Vs: indeed, both In and Bi are routinely used in III–V MBE, and so many existing chambers could support InBi growth without modification. However, thin-film InBi growth is particularly challenging owing to its melting point of 109 °C [38], which is far lower than typical temperatures used for III–V MBE. Heteroepitaxial InBi MBE growth has been attempted by several groups [39,40,41], though achieving a crystalline film has proved to be very difficult. Reflection high-energy electron diffraction (RHEED) patterns are reported to disappear almost instantaneously on InBi deposition by MBE [39]. Film surfaces are prone to droplet formation, and even when the droplets are able to coalesce, the resulting film layer may be discontinuous [39]. The growth of thick layers of tetragonal InBi can be achieved by overpressurization of Bi, increasing the melting point and preventing droplet formation but also increasing the abundance of pure Bi crystallites throughout the film [40]. Ultra-thin films of a hexagonal structure $\sqrt{7}\times \sqrt{7}$ InBi in the 1–2 mL range have been successfully grown with high surface coverage at annealing temperatures greatly exceeding the InBi melting point [41], suggesting that crystalline InBi structures can be stabilized well above the bulk melting point by epitaxial substrate interaction. Interestingly, lattice mismatch did not have a significant effect on the quality of InBi MBE-grown films [39]. Growth on (100) yttria stabilized zirconium (effective mismatch only 3%) reportedly yielded similar film quality to growth on GaAs (effective mismatch 13%). This may relate to the natural cleavage of InBi in its (001) plane, which corresponds to weaker chemical bonding both within the material and across its interface with the substrate, allowing large mismatches to be accommodated in a manner similar to van der Waals epitaxy.

In this work, we explore heteroepitaxial growth of InBi on InSb(001) substrates. InSb has the largest lattice constant of any III–V material, giving a lower lattice mismatch to InBi (9.4%) than GaAs (13.0%) or InP (14.6%). However, as discussed, this may not be significant. The expected heteroepitaxial orientation is shown in Figure 1 as the red square on the left, the InBi unit mesh edges aligning with the <110> directions of InSb(001). An alternative alignment, with the InBi unit mesh edges parallel to [100] and [010] is also shown. The lattice mismatch for that orientation is 22%, but a possible 4:3 mismatch epitaxy would reduce this to an effective 3.1%. Both of these orientations, in fact, appear in this work and depend on sample growth conditions. Oblique orientations with larger minimum common interface areas have been observed for hexagonal-on-cubic epitaxy such as MnAs/GaAs(001) [42] and MnSb/GaAs(001) [43], but consideration of bulk symmetry and dangling bond orientation do not favor such oblique epitaxy in the present case. Other substrates may be useful, such as Si (effective mismatch 7.7%) or SrTiO_2_ (10.5%).

The In-Bi binary phase diagram is shown in Figure 2. At low temperatures, immediately to the In-rich side of the stoichiometric line an In_5_Bi_3_ phase occurs, followed by In_2_Bi. On the Bi-rich side, the InBi phase coexists with Bi, with congruent melting of InBi at 109.2 °C. The five samples discussed below are marked on the phase diagram for context.

## 2. Materials and Methods

Approximately 8 mm-square InSb(001) samples were cut from undoped wafers, which were supplied by Wafer Technology (Milton Keynes, UK), and mounted to flag-style sample plates using either spot-welded Ta wire or indium bonding. After loading to the MBE ultra-high-vacuum (UHV) system, a degassing cycle of 200 °C for at least 30 min was used first. Cycles of argon ion sputtering (500 eV), atomic hydrogen cleaning [45], and annealing at 300 °C were then employed until sharp RHEED patterns were obtained. Crystallographic orientation was achieved by monitoring the Kikuchi features and noting that this surface preparation method produces a mixed (4 × 2) + c(8 × 2) reconstruction, as shown in Figure 3 on row A. In-containing III–Vs are usually somewhat enriched with In at the surface by sputter–anneal cycles [46]. In order to remove excess surface In, samples can be annealed under Sb_4_ flux. This produces either a (1 × 3) or c(4 × 4) reconstruction depending on the annealing duration and temperature. These latter reconstructions are associated with an Sb-terminated surface [47] and are shown in rows B and C of Figure 3. The known InSb surface lattice spacings are shown in C: these were used to calibrate subsequent lattice spacing changes in each sample. Note that the curvature of the RHEED streaks apparent in A [100] and [010] is a result of one-dimensional disorder [48]. The mixed (4 × 2) + c(8 × 2) comprises (4 × 2) sub-units with a disordered arrangement in the ×2 direction. This extends the reciprocal lattice rods into planes, and the intersection of these planes with the Ewald sphere generates curved RHEED streaks.

Samples could be moved in UHV to an adjacent chamber for analysis by X-ray photoelectron spectroscopy (XPS) using non-monochromated Al Kα radiation and a 50 mm mean radius hemispherical analyzer (PSP Vacuum Technology, Macclesfield, UK). All film compositions quoted were calculated from XPS data using standard sensitivity factors. After growth, samples were studied by atomic force microscopy (AFM) in ambient conditions using a standard non-contact mode (Park Systems, Suwon, Republic of Korea).

For epitaxial growth, indium and bismuth were evaporated from shuttered effusion cells, with the beam equivalent pressures (BEPs) calibrated by using a retractable ion gauge. The BEP ratio was used to characterize the growths, along with substrate temperatures. The latter were kept in the ultra-low range, below the melting point of bulk InBi, and were measured by a thermocouple on the sample manipulator. Growth was either done by co-deposition (MBE) or alternate cycles of In and Bi deposition (periodic supply epitaxy, PSE). The MBE samples were grown for a total of 30 min, with BEP values in the mid-10^−7^ mbar range. This gives total film thicknesses of around 100 nm assuming a unity sticking coefficient at the low substrate temperatures. The PSE samples were grown with shorter cycles of In and Bi exposure and at BEP values in the 10^−8^ mbar range in order to investigate the ultra-thin regime. Five samples from the growth campaign are discussed herein, as summarized in Table 1.

## 3. Results

### 3.1. MBE Growth

RHEED patterns for sample 1-A are shown in Figure 4. At the end of InBi growth, an unreconstructed streaky RHEED pattern is observed (row A), although the streaks are less distinct and the background is higher when compared to clean InSb(001). There are also some transmission spots superimposed on the streaky pattern: most notably in the [1$\bar{1}$0] direction. The lattice spacing is consistent with InBi, and the fourfold symmetry suggests (001) orientation. This establishes that epitaxial InBi(001) with reasonable crystallinity can be grown at an ultra-low substrate temperature of 86 °C.

However, the RHEED pattern evolves in a complex way towards this outcome. The progression over time is shown in rows B and C of Figure 4. Starting with a (1 × 3) reconstructed InSb(001) surface [time 0.00], the pattern faded completely within a minute [time 2.00]. The RHEED pattern was very dark, and the image shown in Figure 4 has been slightly contrast-enhanced to allow the shadow edge to be seen. No diffraction features of any kind were visible. This non-crystalline surface recovered to show patterns with transmission diffraction and facet streaks [times 5.30 and 9.30]. This intermediate crystalline phase, hence, comprised 3D nano-islands. The in-plane lattice parameters were 6.43(9) Å and 4.51(6) Å in the [110] and [1$\bar{1}$0] directions, respectively, and 3.31(4) Å and 3.26(4) Å in the [100] and [010] directions, respectively. The out-of-plane lattice constant was 4.86(7) Å. Note that the same structure was also identified in PSE experiments after opening the Bi shutter, which suggests a Bi or Bi-rich phase. The Materials Project database includes a monoclinic Bi structure (mp-1096851) [49] with *a* = 3.24 Å, *b* = 3.22 Å, and *c* = 6.44 Å. The resulting Bi(110) surface of this structure matches well with observed dimensions. Another candidate exists, Bi-III, with a monoclinic cell of dimensions *a* = 4.20 Å, *b* = 4.65 Å, and *c* = 6.65 Å. This has been experimentally recorded at high pressure [50], and here, the pressure may be mimicked by biaxial stress incurred by lattice mismatch. The Bi-III(010) surface does not match the observed dimensions quite as accurately, though the slight compression of the *c* lattice constant may lead to a corresponding expansion of the *a* and *b* parameters.

The film becomes smoother as growth proceeds [time 20:30], but transmission and surface diffraction patterns continue to coexist in the [1$\bar{1}$0] azimuth [time 21:45]. The final disposition of the RHEED [time 30:00] is a coexistence of the streaky pattern consistent with InBi(001) alongside a residual transmission pattern that is assigned to a monoclinic Bi phase. The epitaxial orientation of the InBi should be noted: it lies parallel to InSb [100] and [010] rather than parallel to the primitive surface unit mesh of InSb(001), as shown on the right of Figure 1. The intermediate non-crystalline layer with 3D Bi nano-islands may decouple the epitaxial relationship between InBi and InSb but, nonetheless, maintain sufficient crystallinity that the InBi film forms with a definite orientation. Other samples grown under similar conditions showed the same tendencies, i.e., loss of crystalline order at initial stages and transmission diffraction from Bi 3D nano-islands.

Sample 1-B was grown under more In-rich conditions and was also starting with a (4 × 2) + c(8 × 2) reconstructed InSb(001) surface rather than the Sb-terminated (1 × 3). RHEED patterns before and after growth are shown in Figure 5. After growth, sharp but weak surface diffraction streaks are visible, with little indication of transmission diffraction. The lattice parameters in the [100] and [110] directions deviate considerably from both the expected InBi(001) and the starting InSb(001) parameters. This crystal configuration is close to the In_5_Bi_3_ lattice structure and is in agreement with a=8.54 Å as measured by other researchers [51,52]. The substrate RHEED pattern faded entirely within 1 min of growth, suggesting a period of non-crystalline growth as observed for sample 1-A and other direct MBE samples. Sample 1-B did not exhibit any intermediate crystalline phase and transitioned directly to the In_5_Bi_3_ structure. This is consistent with an In excess on the surface, which would make a coexisting crystalline Bi phase unlikely. The final composition measured by XPS was indeed In-rich but not to the extent that the BEP ratio had been reduced (Table 1).

Sample 1-C was grown under even more Bi-rich conditions. The substrate diffraction pattern rapidly faded soon after initiating deposition and did not return by the end of growth, indicating a non-crystalline film layer was grown. The final Bi:In ratio of 1.23 measured by XPS deviated from InBi stoichiometry. It was found that excess Bi generally caused rapid and permanent loss of crystalline RHEED features.

### 3.2. PSE Growth

Samples 2-A and 2-B were each grown with a sequence of In and Bi shutter operations, but for sample 2-A, the epitaxy was interrupted to allow XPS measurements to be made at each stage, while sample 2-B was grown without interruption and with frequent recording of the RHEED pattern. A summary of the shutter operations for sample 2-A is given in Table 2. Also shown are atomic abundances derived from XPS, the Bi/In ratio derived from both XPS and the time-integrated BEP values, and the observed RHEED spacing with the beam along the substrate [110] direction. Sample XPS spectra are given in Figure 6. These were calculated using shallow core photopeaks, i.e., those of the lowest binding energy and, hence, the highest kinetic energy and effective probing depth, to mask any effects of surface enrichment of either species. However, the films are sufficiently thin that compositions derived from lower-binding-energy photopeaks were in good agreement.

The small excess of In for the as-prepared substrate (Table 2) reflects the generally In-rich nature of the (4 × 2) + c(8 × 2) after sputter cleaning. This provides a small reservoir of In for incorporation into growing InBi in addition to the incident In flux. The InBi films are thin enough that the In reservoir can readily segregate into the growing material and have a notable influence on stoichiometry for the first few layers. Surface enrichment effects may also be inferred from the increase in relative Bi/In film concentrations between the second and third deposition stages (0.65 to 0.92) even though the integrated BEP flux ratio remains almost constant (2.50 to 2.63). This reflects the fact that the surface In reservoir has been “used up”, and so the same incident fluxes produce more Bi-rich material. If growing on an In-rich surface, it is favorable to begin periodic deposition with Bi flux that is adequate to fully incorporate the surface In content. The In:Bi atomic ratio was measured at close to unity by the end of growth.

This stoichiometric evolution is reflected in the shallow core XPS spectra, as shown in Figure 6. In particular, the Bi 5d doublet increases in intensity and sharpens for the second and third growth cycles. The reduction of the Sb 4d peak intensity is consistent with full surface coverage, i.e., the InBi layer is attenuating the XPS signal from the substrate. The In 4d intensity drops as the surface In reservoir is incorporated into the InBi. Figure 6B gives an example survey spectrum showing In, Bi, and Sb peaks without other elements present. The fitting procedure for deriving XPS intensity ratios is illustrated in panel C (shallow core region) and D to F (core levels). A separate Shirley background was applied to each peak (including the wider-spaced spin–orbit doublets), and peak shapes were fitted with single Gaussian–Lorentzian hybrids. Importantly, no chemically shifted components were needed for the InBi, indicating that all of the In and Bi atoms were in the same chemical bonding environment. This further suggests that In-Bi and In-Sb show similar core level binding energies, as would be expected.

RHEED patterns are not shown, for brevity, but we describe the main changes observed. The first stage of growth caused the streaky substrate RHEED pattern to fade completely. It was replaced by a spotty pattern with 3.35(4) Å and 2.35(3) Å d-spacing in the [110] and [100] substrate orientations, respectively. Indium has a body-centered atomic arrangement distorted slightly from BCC to a tetragonal cell with a=3.25 Å and c=4.95 Å, where the former is slightly distorted to match the spacing seen in the [110] RHEED. An expansion of the *a* lattice parameter would produce corresponding compression of the *c* lattice parameter to conserve cell volume, resulting in an expected c=4.66 Å, which agrees with the out-of-plane measurement of 4.69(6) Å. This is a strong indication that a strained In lattice is present with In (001) planes growing parallel to underlying InSb(001) planes. The surface was effectively wetted with a predominantly crystalline In layer. The complete extinction of the existing InSb RHEED pattern is also a good indication that high surface coverage was achieved. The following growth stages returned a similar lattice constant as the substrate after introduction of Bi flux, which incorporates to form a pseudomorphic InBi film. This pattern partially faded after the third deposition cycle but remained at a similar lattice constant, possibly implying a transition to an amorphous structure after a short pseudomorphic growth stage.

A comprehensive view of RHEED evolution is shown for sample 2-B in Figure 7. The first growth cycle comprises 21 s Bi deposition (row A) followed by 24 s In exposure (B). The initial Bi flux lifts the initial InSb(001) reconstruction to form a (3 × 1) reconstructed surface (A). This is distinct from the Sb-terminated InSb(001)-(1 × 3) and reflects the formation of a new Bi-terminated surface phase. Pronounced curvature of the RHEED streaks in the [010] direction indicates one-dimensional disorder in this reconstruction. After the addition of In (row B), the threefold periodicity disappears, but the one-dimensional disorder remains. Lattice parameters remain close to the substrate values, indicating that the InBi film is pseudomorphic, in agreement with observations from other PSE samples. Row C shows a second cycle of Bi, which causes transmission diffraction spots to appear superimposed on surface diffraction streaks. The clear array in the [110] direction provides an out-of-plane measurement of 4.48(5) Å that is somewhat smaller than the expected *c* parameter of 4.78 Å of bulk InBi. The image in the [1$\bar{1}$0] direction appears to show an additional diffraction pattern from a separate structure with apparent in-plane and out-of-plane spacings of 6.44(10) Å and 4.79(9) Å, respectively. This structure is consistent with the monoclinic Bi phase seen on other samples. Further deposition of In (total 50 s) leads to the pattern shown in Figure 7D, with only one structure visible that has lattice constants close to those of the substrate. The transmission spots are also no longer present, suggesting that In planarizes the surface by reacting with the Bi 3D nano-islands to form a smooth film, even at these low temperatures. Planarization by addition of In was seen on several other PSE samples during the growth campaign. The final deposition of Bi shown in Figure 7E slightly increases RHEED pattern intensity and reintroduces the intermediate Bi structure while also producing 3D islands, indicated by the reappearance of transmission spots. Additionally, the measured Bi:In ratio of 1.09 for this sample is very similar to the predicted composition of sample 1-A, while both samples retain a contribution from Bi crystallites amongst the InBi film diffraction in RHEED patterns.

### 3.3. Film Morphology

AFM topographs of three samples are shown in Figure 8: two thick MBE films (In-rich and Bi-rich) and one thin stoichiometric PSE film. The growth mode varies considerably between samples. Sample 1-C has large flat terraces with steep valleys in between, whereas samples 1-B and 2-B show micro- and nano-island formation. This island morphology is consistent with the transmission spots observed in RHEED, as evidenced by the sub-100 nm size of the islands shown in panels F and I. It should be noted that the suppression of the XPS substrate signal intensity (Sb photopeaks) in all cases means that the valleys between the islands in D–F and G–I do not extend to the substrate, i.e., the morphology is film + islands rather than substrate + islands. The globular droplet-like shape of the islands in B is quite striking, with no sign of crystallographic symmetry, as is the very wide range of island sizes from 1 μm to tens of nm (compare D–F). In the case of the MBE samples 1-C and 1-B, the change of the growth morphology is triggered by the drop in the Bi:In flux ratio. Growth in an indium-rich regime evidently leads to droplet formation. The wide range of droplet sizes suggests liquid-like behavior, with Ostwald ripening allowing large droplets to grow. This is consistent with previous work: InBi films have been recorded with greater coalescence at larger Bi:In BEP ratios [40].

Even though sample 2-B is overall Bi-rich, it shows an islanded morphology (Figure 8G–I). However, these islands have a much narrower size distribution, are slightly elongated, and are much more densely packed. This is not consistent with liquid-like droplet formation and Ostwald ripening. It is plausible that these islands relate to the appearance of transmission features during the Bi deposition cycle rather than liquid In-Bi eutectic behavior [53,54].

## 4. Discussion

The direct MBE growth of strain-relaxed InBi can be achieved at substrate temperatures below 100 °C. However, the film passes through a non-crystalline phase and may include excess Bi or In_5_Bi_3_ depending on stoichiometry. These secondary phases are in agreement with the bulk binary phase diagram (Figure 2). A non-crystalline component at the interface and secondary phases could both make these films unsuitable for use in topological spintronics. In contrast, the growth by PSE of thin InBi films on InSb(001) led to the formation of fully strained pseudomorphic InBi(001), though with a non-planar morphology.

Control of the Bi:In flux ratio is evidently very important where the substrate temperature is well below that needed for Bi re-evaporation and self-limiting stoichiometric growth. The formation of 3D nano-islands appears in the In-rich regime and may reflect liquid droplet-like behavior. Conversely, with excess Bi, monoclinic Bi secondary phases were observed. This may promote 3D growth even for PSE, where the In deposition cycle appears to planarize the substrate rather than promote droplet-like growth. Flux ratio control must also account for any reservoir of surface In left by the substrate preparation method.

While this study has explored MBE growth at substrate temperatures below the bulk melting point of InBi, it is not clear that this is necessary to avoid melting of the film if it is stabilized by contact with a crystalline substrate. For example, sample 1-B was above the melting point of In_5_Bi_3_ (Figure 2), but this structure was stable and crystalline on InSb(001). Higher substrate temperatures may allow nucleation of a more crystalline and more stoichiometric InBi epilayer. The viscosity of In-Bi liquid alloys shows complex temperature dependence, with In_2_Bi inclusions affecting the structure and properties [53], while the solid–liquid interfacial energy has been measured for a eutectic In-Bi alloy [54]. Mapping such thermodynamics to the case of In-Bi alloys coexisting with a solid crystalline surface would be very interesting and would have implications for other growth systems, such as the kinetics of a dynamic wetting layer during InAs-GaAs(001) quantum dot formation [55]. Future MBE and PSE growth studies could focus on ramping the substrate temperature during growth to balance crystal quality and phase purity against droplet formation [39].

This work has explored InBi(001) growth on InSb(001) using very low substrate temperatures and PSE. Different strain states and epitaxial orientations have been demonstrated. While this work has concentrated on InSb substrates, the ideas explored here can clearly be extended to other materials, such as GaAs. Refinement of low-temperature epitaxial approaches will allow this very promising topological nodal line material to be incorporated into semimetal–semiconductor heterostructures.

## Figures and Tables

**Figure 1 molecules-29-02825-f001:**
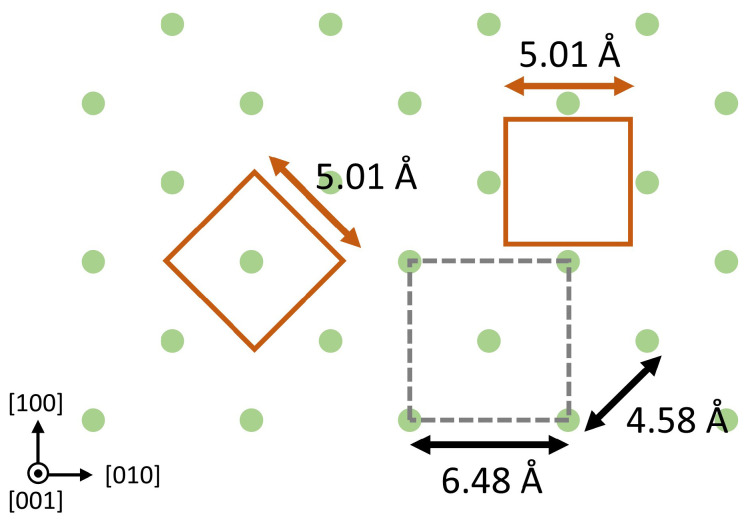
Possible heteroepitaxial orientations for InBi(001) on InSb(001). The InBi unit mesh is shown as red squares.

**Figure 2 molecules-29-02825-f002:**
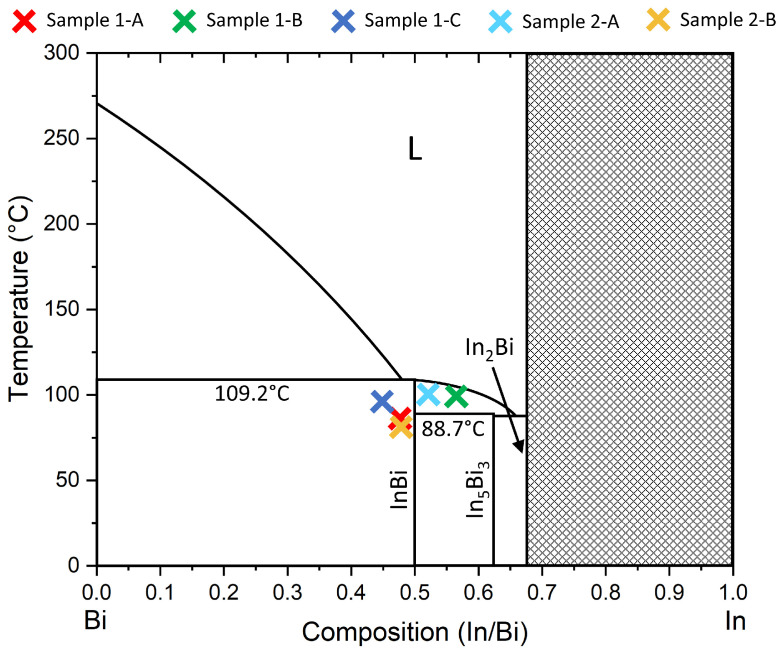
Bulk binary phase diagram for the In-Bi system with approximate locations of the samples discussed herein as colored crosses. Adapted from [38] and using parameters from [44].

**Figure 3 molecules-29-02825-f003:**
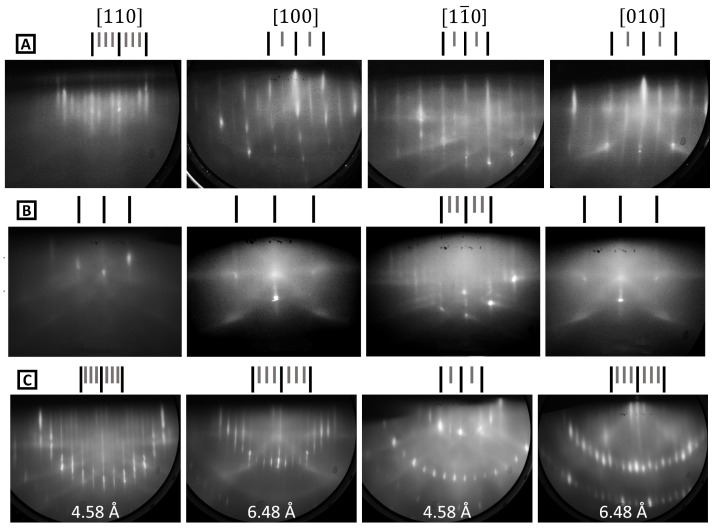
RHEED patterns in the four principal orientations for InSb(001) after (**A**) surface cleaning and (**B**,**C**) additional anneals under Sb_4_ flux. (**A**) represents the mixed (4 × 2) + c(8 × 2), (**B**) is for (1 × 3), and (**C**) is for c(4 × 4). The long and short dashes above the patterns highlight the integer order and the fractional order diffraction rods, respectively. The InSb surface lattice spacings are shown in (**C**).

**Figure 4 molecules-29-02825-f004:**
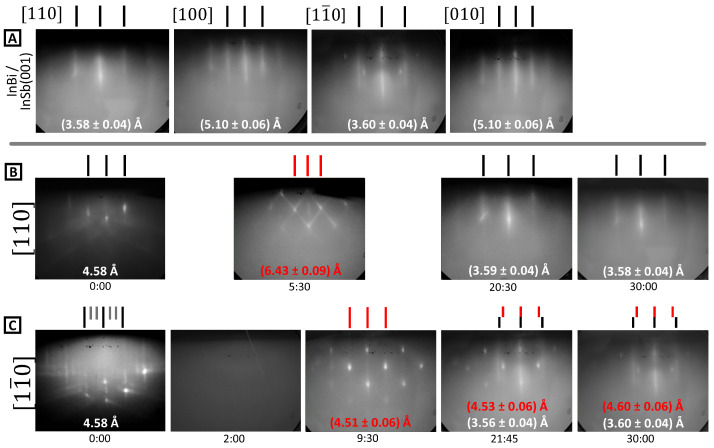
RHEED data for sample 1-A. Row (**A**) shows the four principal directions after growth, with lattice spacing recorded on each image. Rows (**B**,**C**) show patterns as a function of time during growth (in minutes, shown beneath each panel) in the [110] and [1$\bar{1}$0] directions, respectively. Features with a strong transmission diffraction character are labeled in red.

**Figure 5 molecules-29-02825-f005:**
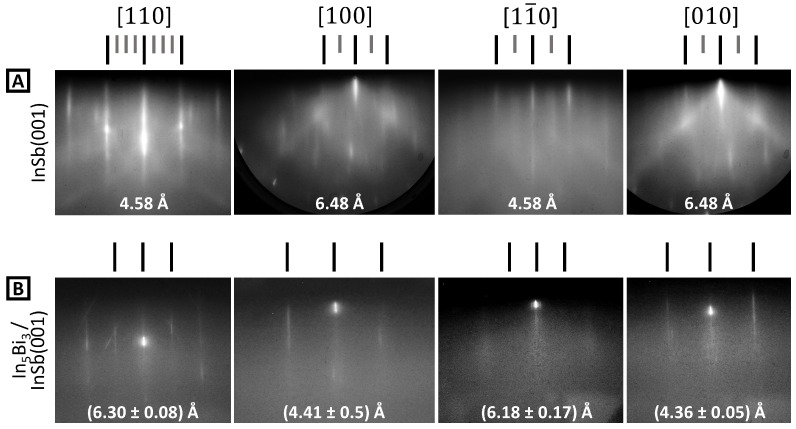
RHEED data for sample 1-B. Row (**A**) shows the four principal directions before growth, with InSb(001) (4 × 2) + c(8 × 2) reconstruction clear. Row (**B**) shows the corresponding patterns after InBi growth.

**Figure 6 molecules-29-02825-f006:**
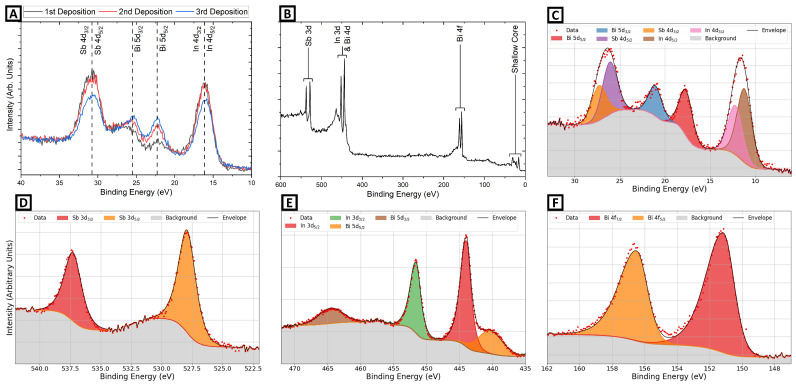
XPS data for sample 2-A. (**A**) The shallow core region at different stages of deposition. (**B**) Survey spectrum after the third deposition cycle. The corresponding region spectra with fitted photopeaks are (**C**) shallow cores, (**D**) Sb 3d, (**E**) In 3d and Bi 4d region, and (**F**) Bi 4f.

**Figure 7 molecules-29-02825-f007:**
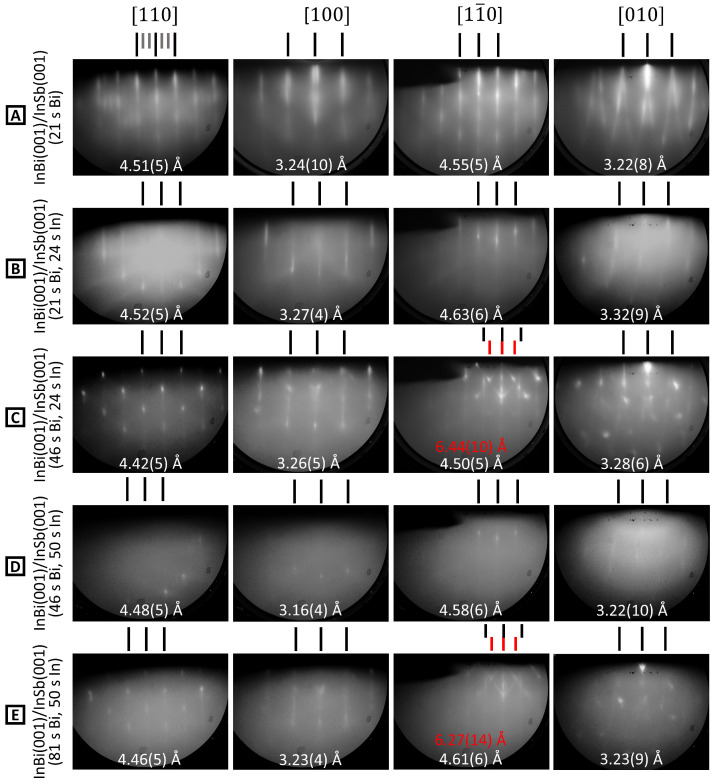
RHEED evolution for sample 2-B (**A**–**E**). The growth stage is indicated on the left, with total Bi and In exposure times noted. The four main azimuthal directions are shown. Surface lattice spacings are shown on each image, and the red value in C [110] indicates an additional structure.

**Figure 8 molecules-29-02825-f008:**
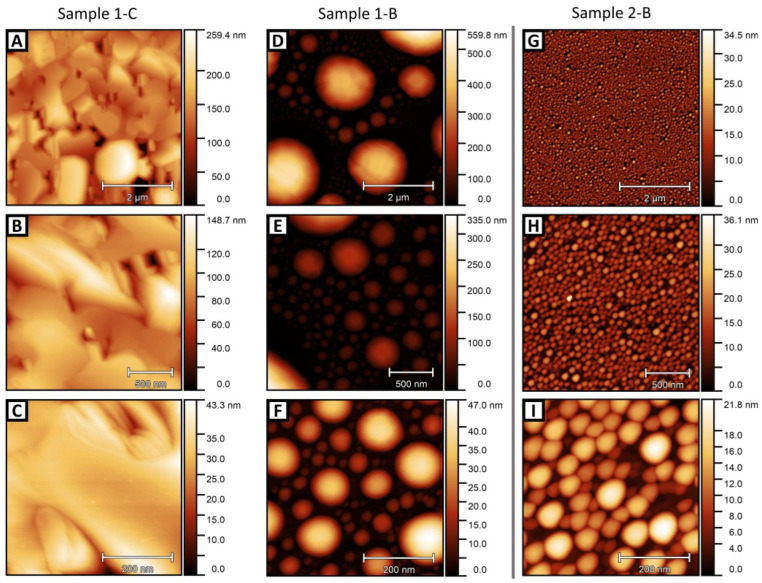
AFM topographs for samples 1-C (**A**–**C**), 1-B (**D**–**F**), and 2-B (**G**–**I**). Rows are ordered by decreasing image size.

**Table 1 molecules-29-02825-t001:** A summary of growth conditions for heteroepitaxial InBi/InSb(001) thick (MBE) and ultra-thin (PSE) film samples. Film compositions based on in situ XPS are also shown. ^a^ XPS measurement could not be performed: composition is based on interpolation of similar samples based on growth conditions.

MBE (thick films)			
Sample	Substrate temp (°C)	Bi:In BEP ratio	Bi:In composition ratio
1-A	86	0.78	∼1.10 ^a^
1-B	99	0.38	0.77
1-C	96	0.94	1.23
**PSE (ultra-thin films)**			
Sample	Substrate temp (°C)	Bi:In Int. BEP ratio	Bi:In composition ratio
2-A	100	2.71	0.92
2-B	81	1.30	1.09

**Table 2 molecules-29-02825-t002:** Shutter operations for sample 2-A. XPS-derived atomic abundances are shown as % with In contribution from the substrate subtracted. The Bi/In ratio shown is derived from both XPS and integrated BEP. The in-plane lattice constant measured in the [110] substrate direction ([100] film direction) is also given.

Growth Stage	In (%)	Bi (%)	Sb (%)	Bi/In XPS	Bi/In BEP	$d_{\left[110\right]}$ (Å)
Pre-growth substrate	55.6	0.0	44.4	–	–	4.58
1st deposition (10s Bi + 30s In)	57.0	6.7	36.3	0.32	0.83	3.35
2nd deposition (20s Bi)	53.6	13.6	32.8	0.65	2.50	4.51
3rd deposition (30s Bi + 30s In)	51.0	22.0	27.0	0.92	2.63	4.57

## Data Availability

The data presented in this study are available on request from the corresponding author.

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
