# Peer review of "Heteroepitaxial Growth of InBi(001)"

_molecules, 2024, doi:10.3390/molecules29122825_

Round 1

Reviewer 1 Report

Comments and Suggestions for Authors This work systematically studies the low-temperature heteroepitaxy growth of InBi on InSb substrates and PSE. The systematic study of low-temperature heteroepitaxy growth provides an effective approach for the preparation of topological materials and semi-metal-semiconductor heterojunctions. This work is very meaningful. I have the following the comments: 1.In Fig.4, the pattern faded completely within a minute [time 2.00], more characterization results or illustration need to be given. 2.For the AFM images corresponding to 1-B and 2-B that are not mentioned in the article, it is better to provide the corresponding images. Comments on the Quality of English Language

English grammar needs a little work. For example: In Page 2, Line 85, "parallel to" shoule be changed into "paralleling to".

Author Response

We are grateful for the rapid and thoughtful response of the reviewer. Changes in the paper are highlighted in red.

  • (1) The RHEED pattern is indeed unusually dark at this point. We have boosted the contrast slightly and added extra description to aid the reader.
  • (2) We have expanded the AFM figure (Fig. 8) from 3 to 9 images to more clearly show the surface morphology at different scales but we did not obtain AFM data for other samples. The discussion is expanded and re-ordered slightly to clarify the differences.

All English usage has been checked (“parallel to” is standard English usage).

Reviewer 2 Report

Comments and Suggestions for Authors

Dear authors,

Thank you for presenting the manuscript 'Heteroepitaxial growth of InBi(001)' to Molecules. It addresses the manifold problems occuring in the process of producing InBi as a thin film material and may therefore be of interest for the community. I recommend the publication of the manuscript after some minor corrections:

(1) In Figure 1 only two very simple possibilities with very obvious misfit (01 and 11) are given for epitaxial orientation of InBi (001) on InSb. Could you include other, more complicated ones? Or could you even give a Table wit hmisfit parameters for orientations such as 21 or 31? Could there be a better substrate material?

(2) Is the melting of InBi at ca. 110 °C comgruent or incongruent? Does InBi have a phase width, however small? Please comment shortly on this.

Sincere greetings

Author Response

We are grateful for the rapid and thoughtful response of the reviewer. Updates to the manuscript are highlighted in red.

  • (1) Consideration of symmetry and dangling bond orientation suggests that these are the two most likely orientations, and indeed these are seen in experiment. Oblique orientations are observed in, for example, hexagonal-on-cubic epitaxy but we are not aware of cubic/tetragonal oblique epitaxy in the III-Vs. We have added a couple of references to oblique epitaxy (MnAs and MnSb) and stated that these are not likely in the present case. We also included mismatches for a couple of other materials in the text.
  • (2) We believe melting is congruent (no phase width is reported) according to published work and have stated this.